# Assistive Prompt Mediation: Evaluating Language Models Under Accessibility Constraints

**Priyaranjan Pattnayak** [1]   **Ishan Banerjee** [2]

## Abstract

Large language models (LLMs) are increasingly used as assistive interfaces for users who cannot reliably produce clean text due to accessibility constraints, yet existing evaluations assume iterative input repair and focus on task accuracy or generic noise robustness. We introduce Assistive Prompt Mediation (APM), a theory-grounded evaluation paradigm that reframes assistance as a constrained mediation problem: recovering latent user intent from accessibility-impaired input without clarification, while minimizing cognitive burden and hallucination risk. APM decomposes assistive quality along these axes and is instantiated across 8 languages, 4 accessibility-driven noise classes, and 10 frontier LLMs, with impairment severity yielding accessibility sensitivity curves. Results show that apparent robustness often masks trade-offs—high intent preservation frequently coincides with increased burden or hallucinated mediation, hallucination rates vary by more than $2\times$ across noise types, and assistive decisions exhibit bounded entropy ($< 0.81$ normalized), indicating systematic rather than unstable behavior. These findings demonstrate that standard robustness metrics substantially overestimate assistive reliability and motivate evaluating LLMs as constrained mediators under accessibility-driven input degradation.

## 1. Introduction

Large language models (LLMs) are increasingly used as assistive interfaces for users who cannot reliably produce clean, well-formed text due to accessibility constraints such as dyslexia, motor impairments, speech recognition errors,

or low vision (Mu et al., 2026; Jain et al., 2020). In these settings, assistance is not simply correcting noisy input, but mediating between accessibility-impaired text and a user's latent intent under constraints on effort, assumptions, and risk. However, most evaluation paradigms assume users can iteratively repair malformed inputs and primarily measure task accuracy or surface-level robustness to noise (Belinkov & Bisk, 2017), providing limited insight into assistive reliability when repair is infeasible.

We introduce *Assistive Prompt Mediation* (APM), an evaluation paradigm that frames assistance as a constrained mediation task. Under accessibility constraints, models must absorb cognitive burden, preserve intent without over-interpretation, and avoid hallucinated assumptions, all without requesting clarification. Given accessibility-impaired input, a model must rewrite the prompt to reflect the user's latent intent while minimizing cognitive burden and hallucination risk. We formalize this setting through three orthogonal dimensions: intent preservation, burden reduction, and hallucination avoidance, defining theory-aligned criteria for reliable assistance. This formulation exposes failure modes overlooked by evaluations centered on unconstrained generation or downstream task completion, where models that appear robust under standard perturbations may still produce burdensome, misleading, or unsafe outputs (Maynez et al., 2020; Mu et al., 2026).

We instantiate APM using 40,960 prompts spanning eight languages and multiple writing systems, collectively representing over five billion native speakers. Accessibility constraints are modeled through four classes of impairment-driven noise (N1–N4) with varying severity, producing *accessibility sensitivity curves* that characterize degradation in assistive behavior and expose failure modes invisible to single-point robustness evaluations.

Across ten frontier LLMs, we find that high intent preservation often co-occurs with increased cognitive burden or hallucinated mediation, revealing a form of false robustness not captured by conventional noise metrics. Despite behavioral divergence across models, assistive decisions exhibit bounded entropy, indicating systematic rather than unstable mediation strategies. Together, these findings show that standard robustness evaluations substantially overestimate

[1]Oracle America Inc., USA [2]Indian Statistical Institute, Bangalore, India. Correspondence to: Priyaranjan Pattnayak <priyaranjanpattnayak@gmail.com>.

*Proceedings of the $43^{rd}$ International Conference on Machine Learning*, Seoul, South Korea. PMLR 306, 2026. Copyright 2026 by the author(s).

assistive reliability and motivate treating language models as constrained mediators under accessibility-driven input degradation.

**Contributions.**

- We introduce Assistive Prompt Mediation (APM), the first evaluation framework to formalize assistive prompt rewriting as constrained mediation under accessibility-impaired inputs.

- We propose theory-aligned metrics capturing intent preservation, cognitive burden, and hallucination risk, enabling diagnosis of assistive failure modes beyond task accuracy.

- We construct a large-scale, multilingual evaluation suite of 40,960 accessibility-impaired prompts spanning eight languages, four assistive task categories, and four impairment-driven noise classes (N1–N4).

- We conduct a comprehensive evaluation of ten frontier LLMs, revealing false robustness and structured divergence in assistive behavior under increasing accessibility constraints.

**Conflict of Interest Disclosure**   The authors declare no financial conflicts of interest related to this work.

## 2. Related Work

**Prompt robustness and perturbation evaluation.**   Several recent works study robustness of LLMs to prompt perturbations or adversarial modifications. For example, PromptRobust introduces a benchmark to measure model resilience to adversarial prompt changes (Zhu et al., 2024). Subsequent methods such as Flip-Flop Consistency propose training approaches to improve performance consistency across prompt variations (Hejabi et al., 2025), and others explore robustness enhancement strategies against typographical or character perturbations in prompting (Mu et al., 2026; Alahmari et al., 2025). These efforts focus on output quality or robustness under variations of otherwise well-formed prompts, whereas APM evaluates assistive mediation from accessibility-impaired inputs that cannot be repaired, scoring constrained rewriting behavior rather than downstream task utility.

**Prompt rewriting and query reformulation.**   Prompt rewriting has been explored as a mechanism to improve downstream task performance by paraphrasing or optimizing user prompts (Shin et al., 2020; Pryzant et al., 2023). These approaches typically evaluate success via answer quality and allow unconstrained rewriting or iterative refinement.

In contrast, APM treats rewriting itself as the assistive output and evaluates whether the rewritten prompt faithfully recovers user intent while minimizing cognitive burden and unsupported assumptions, without allowing clarification. As such, prompt-rewriting methods are not directly comparable, as they optimize task utility rather than constrained assistive mediation under accessibility-impaired inputs.

**Accessibility and disability evaluation.**   Recent work has evaluated LLM behavior with respect to disability-related bias, tone, or framing using paired neutral and disability-aware prompts (e.g., AccessEval) (Panda et al., 2025). This line of work studies disparities in responses conditioned on disability mentions. APM addresses a complementary but distinct problem: accessibility-impaired interaction, where the input itself is degraded by impairment and the model must mediate intent under constraint, rather than respond to explicit disability cues.

**Multilingual evaluation under degraded inputs.**   Multilingual benchmarks such as XTREME and FLORES assess cross-lingual generalization under clean, well-formed inputs (Hu et al., 2020; Goyal et al., 2022). APM instead treats language and script as interacting stressors under accessibility-driven degradation, evaluating how assistive mediation behavior varies across languages when user input quality deteriorates.

**Hallucination and assistive reliability.**   Hallucination and faithfulness have been studied extensively, often defined as factual inconsistency with a grounding source (Maynez et al., 2020; Ji et al., 2023). APM reframes hallucination as an assistive failure mode: unsupported intent additions introduced during mediation, even in the absence of external grounding. This reflects assistive risk under ambiguity rather than factual error alone.

In summary, prior work addresses robustness, prompt rewriting, accessibility, or multilinguality in isolation. APM integrates these threads under a unified assistive mediation task, evaluating constrained behavior under non-repairable accessibility impairments, a setting not captured by existing benchmarks or evaluations.

## 3. Dataset

### 3.1. Assistive Prompt Set

Assistive Prompt Mediation (APM) targets interactions that are not represented in existing NLP benchmarks, which typically assume clean or repairable user input and allow clarification-based repair. In contrast, APM models scenarios in which users are structurally unable to produce well-formed prompts and cannot participate in iterative clarification. Prompts must therefore encode latent assistive

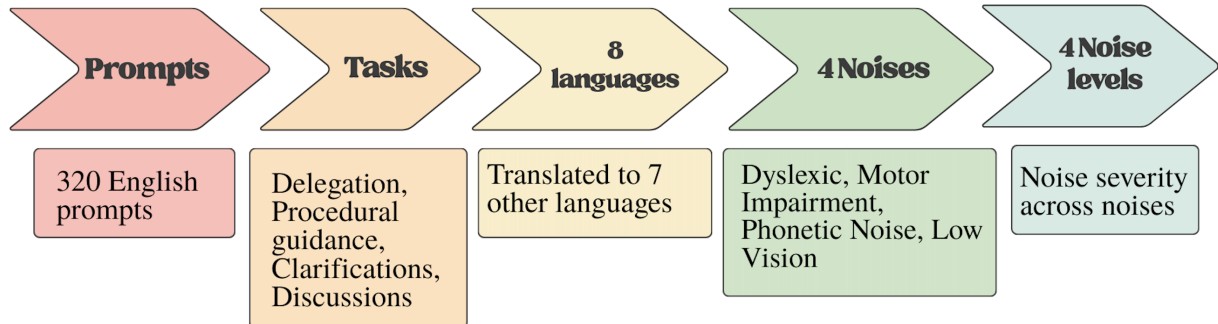

*Figure 1.* Overview of Assistive Prompt Mediation dataset showing categories, multilingual noise types and severity.

intent rather than be adapted from task-oriented datasets.

We construct a custom prompt set to serve as a controlled evaluation substrate for assistive mediation. The ASSISTIVE PROMPTING DISABILITIES DATASET[1] consists of 320 clean prompts authored in English, each expressing a single, unambiguous assistive intent and requiring no external tools or domain-specific knowledge. Prompts are designed to admit objective or judgment-based evaluation of assistive utility, ensuring that observed differences reflect mediation behavior rather than dataset artifacts.

Prompts are distributed across four assistive intent categories grounded in accessibility research:

- **Communication & Task Delegation**: everyday messages, scheduling requests, and routine assistance tasks;

- **Procedural Guidance & Safety**: stepwise instructions, how-to requests, and risk-sensitive guidance;

- **Cognitive Load Reduction**: requests for simplification, explanation, or rephrasing of information;

- **Decision Support & Comparison**: lightweight reasoning tasks involving choices or comparisons.

These categories reflect documented usage patterns in *Augmentative and Alternative Communication (AAC)* and *Universal Design for Learning (UDL)*, rather than conventional NLP task taxonomies. Prompt intents are language-agnostic by design and were iteratively reviewed to ensure clarity and minimal ambiguity.

### 3.2. Languages

We instantiate the prompt set across eight languages selected for orthographic and structural diversity: English

---

[1]Available at: https://huggingface.co/datasets/Ishan3141/Assistive_Prompting_Disabilities_Dataset.

(en), Spanish (es), Italian (it), German (de), French (fr), Hindi (hi), Chinese (zh), and Japanese (ja). Languages are not treated as coverage targets but as structured stressors for analyzing how accessibility-driven noise interacts with script and writing system properties.

Clean English prompts are translated using a high-quality machine translation system (Google Translate) and treated strictly as a preprocessing step. To verify semantic fidelity, 10% of translated prompts are randomly sampled for back-translation and bilingual human inspection, yielding substantial agreement (Cohen's $\kappa \approx 0.81$) on preserved intent. Translation quality itself is not evaluated or reported.

### 3.3. Accessibility-Driven Noise

Each clean prompt is transformed using four classes of accessibility-driven noise (N1–N4), corresponding to increasing severity of real-world accessibility constraints arising from dyslexia, motor impairments, speech recognition errors, or low-vision text entry. These transformations are non-adversarial and model degraded input that users cannot reliably repair.

Each noise class is parameterized by a severity coefficient $\alpha \in \{0.2, 0.4, 0.6, 0.8\}$, producing graded impairment levels. Sampling multiple severity values enables the construction of *accessibility sensitivity curves*, which characterize how assistive behavior degrades as accessibility constraints intensify.

Across 320 base prompts, eight languages, four noise classes, and four severity levels per noise class, the dataset comprises 40,960 prompt instances. A full breakdown is provided in Appendix A.

## 4. Methodology

APM evaluates large language models as *fixed assistive mediators* operating under accessibility constraints. The methodology therefore specifies (i) how accessibility-driven noise is instantiated from abstract dataset definitions, and

(ii) how assistive mediation is elicited and evaluated under a constrained, non-interactive protocol. No model training, fine-tuning, or adaptation is performed.

### 4.1. Noise Realization and Parameterization

The dataset defines four accessibility-driven noise classes (N1–N4), corresponding to increasing severity of real-world accessibility constraints. Each class is instantiated through deterministic, language-aware transformations that approximate characteristic error patterns in accessibility-impaired text entry while preserving plausible human input trajectories:

- **N1 (Orthographic degradation)**: character-level swaps, deletions, insertions, and duplications reflecting spelling errors common in dyslexia and low-precision text entry (Rello et al., 2017);

- **N2 (Telegraphic structure)**: omission of function words and syntactic compression, modeling AAC-style or motor-constrained communication (McCoy, 1997);

- **N3 (Phonetic distortion)**: phonetic spellings, abbreviations, and controlled phoneme confusions approximating ASR and noisy-channel text errors (Wang & Ng, 2013);

- **N4 (Severe truncation)**: removal of contiguous text spans, yielding fragmented input with sparse lexical cues, reflecting partial observability and compounded accessibility constraints.

While surface realizations vary across scripts and orthographies, the underlying noise logic is shared across languages. We do not model cultural, sociolinguistic, or adversarial perturbations beyond their interaction with accessibility constraints.

Each noise type is parameterized by a severity coefficient

$$\alpha \in \{0.2, 0.4, 0.6, 0.8\},$$

where increasing $\alpha$ monotonically increases degradation severity. This parameterization enables controlled stress-testing and supports the construction of *accessibility sensitivity curves*, which characterize how assistive behavior degrades as accessibility constraints intensify.

### 4.2. Assistive Mediation Protocol

All models are evaluated under an identical, frozen assistive mediation instruction that requires rewriting accessibility-impaired input to reflect the user's latent intent while prohibiting clarification requests, unsupported assumptions, or extraneous explanation. The protocol enforces burden absorption by the model and treats deviations from constrained

mediation, such as asking clarifying questions or adding new content as assistive failures.

The full instruction text is provided in Appendix B.

## 5. Evaluation Metrics

APM evaluates large language models as *assistive mediators* operating under accessibility constraints, rather than as task solvers. Metrics are therefore organized around four diagnostic themes that capture distinct assistive failure modes. Table 1 summarizes all metrics and their diagnostic roles.

### 5.1. Semantic Reliability

**Intent Preservation (I, IPR).** Semantic reliability measures whether mediation preserves the user's latent intent. An LLM-based judge compares the clean prompt and the mediated prompt, assigning an ordinal score from 1 (intent lost or incorrect) to 5 (fully preserved intent), ignoring stylistic differences. We report the average intent preservation score (IPR) as the primary indicator of semantic fidelity under accessibility constraints.

### 5.2. Burden Transfer and Usability

A core goal of assistive mediation is to absorb cognitive burden on behalf of the user. We explicitly measure whether mediation reduces or inflates the effort required to interpret the prompt.

**Cognitive Burden Computation.** We use the term *cognitive burden* to denote a structural proxy for usability difficulty, rather than a direct measurement of human cognition. The objective is to quantify properties of mediated outputs that increase processing effort under accessibility constraints, such as verbosity, structural fragmentation, and irregular formatting.

Given a prompt $x$, the burden score is computed as:

$$B(x) = 0.4 \cdot L(x) + 0.3 \cdot \big(P(x) \times 100\big) + 0.3 \cdot \big(H(x) \times 10\big),$$

where:

- $L(x)$ is the token length of the prompt,

- $P(x)$ is the punctuation density, defined as the fraction of punctuation tokens relative to total tokens,

- $H(x)$ is the character-level Shannon entropy of the text.

Tokenization is performed using a lightweight regex-based tokenizer that separates word and punctuation tokens. Entropy is computed over character frequencies:

$$H(x) = -\sum_{c \in \mathcal{C}} p(c) \log_2 p(c),$$

where $\mathcal{C}$ denotes the set of characters in the prompt and $p(c)$ is the empirical frequency of character $c$.

Intuitively, longer outputs increase reading effort, high punctuation density reflects fragmented or structurally complex text, and higher entropy captures lexical irregularity and variability. The scaling constants are used only to place the three components on comparable numerical ranges.

This metric is intended as a deterministic structural proxy for readability-related processing difficulty and is applied identically to both raw and mediated prompts. The resulting scores are then used to compute Assistive Burden Inflation (ABI) and Burden Robustness Score (BRS). While simplified, this formulation enables consistent large-scale comparison across models, languages, and impairment severities without affecting the qualitative conclusions of the study.

**Assistive Burden Inflation (ABI).**   Assistive Burden Inflation measures the expected change in cognitive burden introduced by mediation:

$$\text{ABI} = \mathbb{E}[B_{\text{assist}} - B_{\text{raw}}].$$

Positive ABI indicates usability regressions despite apparent semantic correctness.

**Burden Robustness Score (BRS).**   BRS captures the relative improvement in usability, calculated as the expected reduction in cognitive burden from the raw input to the assisted prompt.

$$\text{BRS} = \mathbb{E}[B_{\text{raw}} - B_{\text{assist}}].$$

Variance in BRS across impairment severities is used to assess stability under increasing accessibility degradation.

Although ABI and BRS derive from the same underlying burden-transfer formulation, they are retained as complementary diagnostics rather than independent measurements. ABI emphasizes relative burden inflation introduced by mediation, making it useful for identifying assistive regressions and false robustness, while BRS emphasizes the absolute usability of the mediated output under accessibility constraints. This distinction mirrors common practice in robustness and reliability evaluation, where sign-related metrics are often reported separately to highlight different operational interpretations.

### 5.3. Assistive Risk and Overreach

Beyond preserving intent, assistive mediation must avoid introducing unsupported assumptions, unnecessary elaboration, or interactional violations.

**Hallucinated Mediation.**   We record whether the mediator introduces unsupported content via a binary hallucination indicator, optionally accompanied by an ordinal severity score to distinguish benign drift from potentially harmful over-interpretation. We report the Hallucination Rate (HIR) as the fraction of mediated outputs containing such additions.

**Protocol Violations.**   Protocol Compliance Rate (PCR) measures adherence to the non-interactive mediation contract, flagging outputs that request clarification, introduce meta-commentary, or add explanations. We further track Over-Assist Rate (OAR) and Clarification Propensity (CP) as indicators of unnecessary intervention, even when intent appears preserved.

### 5.4. Stability under Accessibility Stress

Finally, APM evaluates whether assistive behavior remains structured and consistent as accessibility constraints intensify.

**Ambiguity Entropy (AE).**   Ambiguity Entropy quantifies behavioral unpredictability by computing entropy over discrete assistive outcomes (hallucination, over-assistance, clarification). Lower entropy indicates systematic mediation, while higher entropy reflects brittle or unstable behavior.

**Cross-Condition Agreement & Language Consistency.** We assess consistency via exact-match agreement and normalized entropy over intent preservation scores across (i) languages for the same underlying prompt and (ii) models under identical conditions. These analyses distinguish structured divergence from stochastic instability. To isolate multilingual stability, we introduce the *Language Consistency Index (LCI)*, which measures within-model agreement of assistive behavior across languages for the same underlying prompt. It quantifies whether mediation decisions remain invariant under translation.

**Accessibility Sensitivity Curves (ASC).**   For any evaluation metric $M$, we define the accessibility sensitivity curve as

$$\text{ASC}_M(\alpha) = \mathbb{E}[M \mid \alpha],$$

where $\alpha$ denotes impairment severity. In addition to standard fidelity and burden metrics, we report ASC with *Assistive Success Rate (ASR)* as the response variable. Assistive success is defined as the simultaneous satisfaction of protocol compliance and intent preservation, with neither hallucinated mediation nor cognitive burden inflation. For an interaction $i$,

$$\text{AS}_i = \mathbb{I}[P_i]\,\mathbb{I}[I_i]\,\mathbb{I}[\neg H_i]\,\mathbb{I}[\neg B_i],$$

*Table 1.* APM evaluation metrics grouped by four core assistive themes and their diagnostic sub-metrics.

| Assistive Theme | Metric(s) | Interpretation |
|---|---|---|
| Semantic Reliability | Intent Preservation (I, IPR) | Fidelity of mediated prompt to the user's latent intent under accessibility-impaired input. |
| Burden Transfer and Usability | Cognitive Burden (B), ABI, BRS | Whether mediation reduces user effort and remains usable as impairment severity increases. |
| Assistive Risk and Overreach | Hallucination Rate (HIR), Severity, PCR, CP, OAR | Introduction of unsupported content or violations of the constrained mediation protocol. |
| Stability under Accessibility Stress | Ambiguity Entropy (AE), LCI, ASC | Consistency and degradation of assistive behavior across severity and languages. |

where $P_i$ denotes protocol compliance, $I_i$ intent preservation, $H_i$ hallucinated mediation, $B_i$ burden inflation and $\neg$ is the "not" operator. The corresponding accessibility sensitivity curve is then

$$\text{ASC}_{\text{ASR}}(\alpha) = \mathbb{E}[\text{AS}_i \mid \alpha].$$

Table 1 summarizes the evaluation themes and corresponding metrics used to characterize assistive mediation behavior under accessibility constraints.

### 5.5. LLM-Based Judging Protocol

Several assistive metrics require semantic and interactional judgments that are not deterministically computable at scale. Specifically, *Intent Preservation (IPR)*, *Hallucination Incidence Rate (HIR)*, and protocol-related metrics (*PCR*, *CP*, *OAR*) are computed using an LLM-based judge.

We use GPT-5.2 as the judge model with a fixed evaluation prompt that compares the clean reference prompt, the accessibility-impaired input, and the mediated output. IPR is the mean of the intent_preservation score, HIR is the fraction of outputs with hallucinated_additions = true, and LCI is computed from agreement on language_match across translations of the same prompt. The judge prompt is fixed across all experiments and is provided in Appendix D. To validate judge reliability, we randomly sampled 5% of evaluated outputs across models, languages, noise classes, and severity levels. Two independent human annotators re-labeled these samples using the same criteria. Agreement with the LLM judge was high: Spearman $\rho = 0.82$ for ordinal intent preservation scores, quadratic-weighted Cohen's $\kappa = 0.79$ for intent preservation and $\kappa = 0.72$ for hallucination severity, and Cohen's $\kappa = 0.76$ for binary hallucination and protocol compliance decisions.

To assess robustness against judge-specific bias, we additionally evaluated a full evaluation using Gemini 3 Pro as an independent judge. Rankings and aggregate trends remained highly correlated ($\rho \approx 0.81$–$0.86$ across metrics), indicating that conclusions are not dependent on a single LLM judge.

## 6. Results

We evaluate Assistive Prompt Mediation (APM) across 10 frontier LLMs, 8 languages, 4 accessibility-driven noise classes (N1–N4), and 4 impairment severities ($\alpha \in \{0.2, 0.4, 0.6, 0.8\}$), totaling 40,960 mediated prompts per model. We organize results to mirror the evaluation themes in Section 5: (i) semantic reliability, (ii) burden transfer and usability, (iii) assistive risk and overreach, and (iv) stability under accessibility stress.

### 6.1. Semantic Reliability: Intent Preservation (IPR)

We evaluate semantic reliability using Intent Preservation (IPR), which measures whether mediation preserves the user's latent intent (Section 5). Under mild impairment ($\alpha = 0.2$), all models achieve high intent preservation (mean IPR $\geq 4$), but intent degrades sharply under severe impairment ($\alpha \geq 0.6$), as shown by the Accessibility Sensitivity Curves (ASC) in Figure 2. Model rankings are not stable across severity: while several models perform well at $\alpha \leq 0.4$, differences collapse at higher severities, where no model consistently preserves intent across noise types. This shows that robustness measured under mild accessibility noise substantially overestimates semantic reliability in realistic assistive settings.

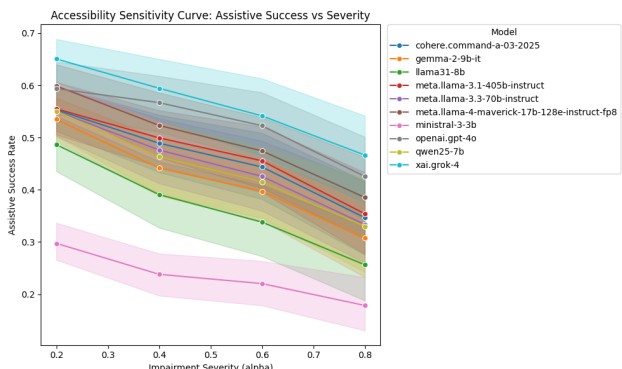

*Figure 2.* Accessibility Sensitivity Curve for various models

## 6.2. Burden Transfer & Usability: BRS, ABI, and False Robustness

We next assess whether mediation *reduces* cognitive burden or inadvertently inflates it (BRS, ABI; Section 5). Table 2 summarizes average Burden Robustness Score (BRS) by noise class and severity, showing monotonic usability degradation with increasing $\alpha$, with N2 inducing the largest burden increases.

*Table 2.* Heatmap-style Average Burden Robustness Score (BRS) across impairment severity. Higher BRS is better; negative values indicate burden inflation. Darker cells indicate better values.

| Noise | $\alpha=0.2$ | 0.4 | 0.6 | 0.8 |
|---|---|---|---|---|
| N1 | 0.04 | -0.20 | -0.21 | -0.39 |
| N2 | -5.81 | -6.42 | -6.91 | -7.34 |
| N3 | -1.02 | -1.48 | -1.97 | -2.31 |
| N4 | -0.62 | -0.94 | -1.21 | -1.53 |

To quantify when assistive failures emerge, Table 3 reports the median impairment severity at which key metrics cross failure thresholds. Notably, usability- and consistency-related metrics (ABI, LCI) fail at lower impairment levels ($\alpha \approx 0.4$) than intent preservation ($\alpha \approx 0.6$), indicating that assistive breakdown often precedes semantic failure.

*Table 3.* Impairment severity ($\alpha$) at which major assistive metrics cross failure thresholds.

| Metric | Threshold | Median $\alpha$ |
|---|---|---|
| Intent Preservation (IPR) | $< 3.5$ | 0.6 |
| Assistive Burden Inflation (ABI) | $> 0$ | 0.4 |
| Hallucination Rate (HIR) | $> 0.10$ | 0.6 |
| Language Consistency (LCI) | $< 0.2$ | 0.4 |

**False Robustness.** False robustness is not a standalone metric but a derived failure mode that emerges from the joint behavior of Intent Preservation (IPR) and Assistive Burden Inflation (ABI): cases where mediation preserves intent (high IPR) while increasing cognitive burden (ABI $> 0$). Figure 4 shows a systematic decoupling between semantic fidelity and usability: many model–condition pairs achieve high IPR yet exhibit burden inflation. Table 4 confirms that even the strongest models display positive ABI alongside high intent preservation, demonstrating that semantic correctness alone is insufficient for assistive quality.

To localize false robustness across stressors, Figure 3 visualizes the false robustness rate (IPR $\geq 4$ and ABI $> 0$) by language and noise type. Consistent with the burden-transfer view, false robustness is driven primarily by noise class (N2/N3) rather than language identity.

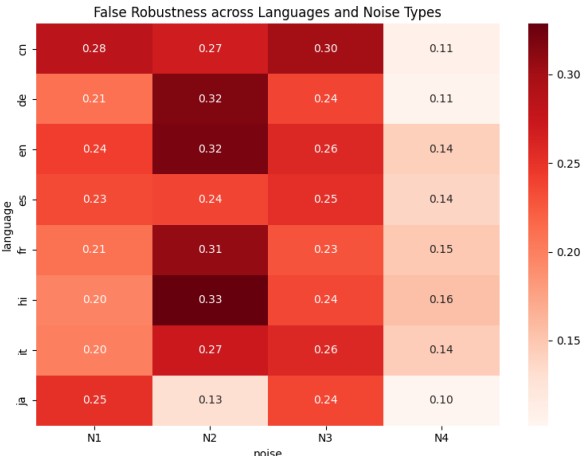

*Figure 3.* False robustness rate across languages and accessibility-driven noise classes. Each cell shows the fraction of cases with high intent preservation (IPR $\geq 4$) but positive Assistive Burden Inflation (ABI $> 0$). Mid-severity noise types (N2, N3) consistently induce higher false robustness across all languages.

## 6.3. Assistive Risk & Overreach: HIR, PCR, CP, OAR

We next evaluate assistive risk and protocol adherence using Hallucination Incidence Rate (HIR) and related overreach metrics. Table 6 reports language-wise aggregates across all models, noise classes, and severities. While hallucination rates vary by language, they are non-trivial even in English (HIR = 27.2%) and higher in several non-English languages such as Hindi (39.6%) and Italian (36.7%).

Hallucination incidence depends strongly on the type of accessibility degradation. Orthographic noise (N1) produces high absolute hallucination rates (e.g., 25.8% in English and 57.0% in Hindi), while mid-severity, semantics-preserving noise (N2/N3) yields hallucination rates ranging from approximately 7–45% across languages (Appendix A, Table 14). Although N1 exhibits higher absolute hallucination, N2 and N3 are more likely to produce hallucinations in conjunction with high intent preservation, contributing disproportionately to false robustness failures.

Table 5 summarizes qualitative assistive failure signatures by noise type. Mid-severity degradations (N2, N3) are characterized by burden inflation and hallucinated mediation despite preserved surface semantics, whereas truncation (N4) primarily reduces semantic reliability and yields conservative rewrites.

Table 6 shows substantial variation in hallucination incidence across languages, with particularly high rates in Hindi (39.6%) and Italian (36.7%), while English and East Asian languages remain non-trivially affected.

*Table 4.* Heatmap-style per-model assistive mediation performance aggregated across all languages, noise classes, and severities. Higher IPR, BRS, and PCR are better; lower HIR is better. Darker cells indicate better column-wise values, accounting for each metric's direction.

| Model | IPR ↑ | BRS ↑ | HIR ↓ | PCR ↑ |
|---|---|---|---|---|
| xai.grok-4 | 4.088 | -2.222 | 0.194 | 0.703 |
| openai.gpt-4o | 4.052 | -1.867 | 0.183 | 0.687 |
| cohere.command-a-03-2025 | 3.881 | -2.449 | 0.293 | 0.698 |
| meta.llama-4-maverick-17b-128e | 3.869 | -1.956 | 0.264 | 0.754 |
| meta.llama-3.1-405b | 3.834 | -1.821 | 0.294 | 0.732 |
| meta.llama-3.3-70b | 3.717 | -1.607 | 0.327 | 0.743 |
| gemma-2-9b-it | 3.573 | -1.196 | 0.288 | 0.714 |
| qwen25-7b | 3.419 | -0.732 | 0.299 | 0.777 |
| llama31-8b | 3.209 | -1.377 | 0.452 | 0.777 |
| ministral-3-3b | 2.563 | -1.600 | 0.513 | 0.826 |

*Table 5.* Noise-specific assistive failure profiles aggregated across all models, languages, and severities.

| Noise | Avg. IPR ↑ | ABI ↓ | HIR ↓ | OAR ↓ | AE ↓ |
|---|---|---|---|---|---|
| N1 (Orthographic) | High | Low | Low | Low | 0.41 |
| N2 (Telegraphic) | High | **High** | **High** | Medium | 0.52 |
| N3 (Phonetic) | Medium | **High** | **High** | Medium | 0.49 |
| N4 (Truncation) | Low | Low | Low | Low | 0.38 |

### 6.4. Stability under Accessibility Stress: AE, LCI, and Agreement

Finally, we assess if assistive behavior remains structured and consistent under increasing accessibility stress. Across all models and conditions, normalized ambiguity entropy remains bounded (AE < 0.81), indicating systematic rather than stochastic mediation behavior even under severe impairment.

Despite this stability, cross-language exact agreement (Table 7) remains low (0.06–0.30), indicating that mediated outputs are rarely identical across translations of the same prompt. Cross-model agreement is even lower (0.06–0.15), reflecting substantial variability in how different models rewrite accessibility-impaired input. However, LCI values remain high across languages (Table 6; mean ≈0.97), indicating that while surface realizations differ, models apply consistent assistive strategies rather than behaving erratically.

Taken together with the Accessibility Sensitivity Curves in Figure 2, these results show that degradation under accessibility stress is structured, predictable, and driven by specific interactions between noise type, severity, and assistive overreach rather than random model failure.

### 6.5. Summary of Quantitative Trends

Across all experiments, we observe that: (i) semantic reliability (IPR) degrades smoothly with severity, but usability and multilingual reliability degrade earlier; (ii) high intent preservation frequently co-occurs with positive ABI, exposing false robustness; (iii) assistive risk (HIR/OAR) is

concentrated in N2/N3; and (iv) assistive behavior exhibits structured divergence across models and languages despite bounded entropy.

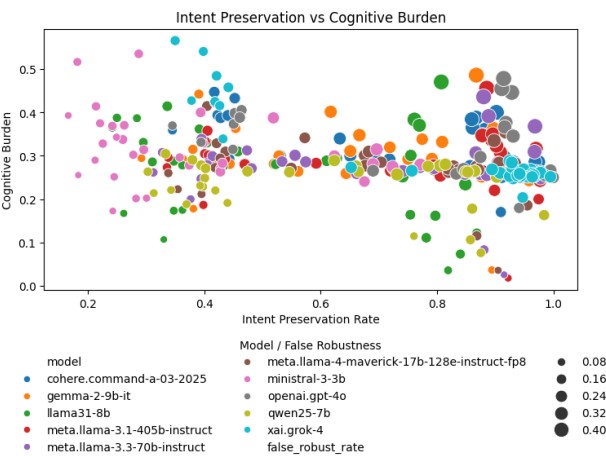

*Figure 4.* Intent preservation vs Cognitive Burden

## 7. Discussion

This study evaluates robustness under accessibility constraints by treating large language models as *assistive mediators* rather than task solvers. Our results reveal a systematic mismatch between semantic correctness and assistive reliability that is invisible to intent-only evaluation.

*Table 6.* Heatmap-style language-wise assistive reliability aggregated across all models, noise classes, and impairment severities. Lower HIR, and Language Drift are preferred; higher BRS (less negative) and LCI are better. Darker cells indicate better value.

| Language | BRS ↑ | HIR ↓ | LCI ↑ | Lang. Drift ↓ |
|---|---|---|---|---|
| English (en) | -1.51 | 0.272 | 0.998 | 0.000 |
| Spanish (es) | -1.26 | 0.334 | 0.945 | 0.452 |
| Italian (it) | -1.35 | 0.367 | 0.932 | 0.491 |
| German (de) | -1.96 | 0.366 | 0.960 | 0.834 |
| French (fr) | -1.34 | 0.360 | 0.953 | 0.873 |
| Hindi (hi) | -0.02 | 0.396 | 0.989 | 2.053 |
| Chinese (zh) | -2.63 | 0.193 | 0.984 | 2.638 |
| Japanese (ja) | -3.39 | 0.200 | 0.983 | 2.445 |

*Table 7.* Cross-language and cross-model consistency metrics averaged across all noise classes and impairment severities. Ranges indicate minimum–maximum values

| Metric | Exact Match | Normalized Entropy |
|---|---|---|
| Cross-language agreement | 0.06–0.30 | 0.73–0.84 |
| Cross-model agreement | 0.06–0.15 | 0.70–0.76 |

**Assistive failure precedes semantic failure.** Across models, usability and consistency degrade at lower impairment levels than intent preservation. Assistive Burden Inflation (ABI) and LCI cross failure thresholds at $\alpha \approx 0.4$, while intent preservation degrades later at $\alpha \approx 0.6$ (Table 3). This ordering shows that mediation often becomes unusable or unreliable *before* semantic intent is lost, contradicting robustness claims based solely on correctness.

**False robustness arises from over-interpretation under ambiguity.** The dominant failure mode occurs when models preserve intent while increasing cognitive burden. This *false robustness* is driven by noise regimes that retain partial semantic cues, not by model scale or language. Mid-severity degradations (N2/N3) consistently induce burden inflation and hallucination with high intent preservation, accounting for most false robustness failures across languages (Figure 3). Thus, partial evidence encourages confident over-elaboration rather than conservative mediation.

**Stability does not imply safety.** Although cross-language exact agreement is low (0.06–0.30), ambiguity entropy remains bounded (0.70–0.84), indicating structured rather than stochastic behavior. High LCI values further show that models apply consistent mediation strategies across languages, even when those strategies produce hallucination or excess burden. Consistency therefore reflects strategy reuse, not assistive correctness.

**Implications for evaluation.** Together, these findings show that semantic fidelity alone is insufficient for eval-uating assistive systems. Robustness under accessibility constraints is governed by interactions between ambiguity, over-interpretation, and burden transfer. Metrics such as ABI, BRS, and LCI are necessary to detect assistive failures that remain hidden under intent-preservation benchmarks, in multilingual and accessibility-critical settings.

## 8. Conclusion

We presented Assistive Prompt Mediation (APM), an evaluation framework for measuring how large language models behave as assistive mediators under accessibility constraints. Across 10 models, 8 languages, and multiple impairment regimes, APM shows that assistive failures emerge well before semantic failure: cognitive burden inflation, hallucinated mediation, and multilingual inconsistency appear at substantially lower impairment levels than intent loss. In particular, mid-severity, semantics-preserving degradations induce a prevalent failure mode—*false robustness*—in which models preserve intent while increasing user effort or introducing unsupported content. These results expose a fundamental limitation of intent-centric robustness evaluation: semantic correctness and behavioral consistency do not guarantee assistive reliability, underscoring the need for evaluation frameworks that explicitly measure usability, risk, and stability in accessibility-sensitive settings.

## Impact Statement

This paper evaluates large language models as assistive mediators for users who experience accessibility-related input impairments, including spelling errors, phonetic substitutions, truncated input, and reduced linguistic structure. Such conditions commonly arise for users with motor impairments, speech or language disorders, cognitive fatigue, low literacy, or when interacting with assistive input technologies.

By systematically analyzing how models preserve intent, transfer cognitive burden, and introduce unintended assumptions under these conditions, this work aims to improve the

evaluation of assistive AI systems before deployment to disabled users. Our findings identify failure modes, such as burden inflation and hallucinated mediation, that could disproportionately affect users who rely on AI systems for communication support, highlighting the need for rigorous assistive evaluation rather than intent-only robustness checks.

This work does not propose new assistive interventions or deployable systems; instead, it provides an evaluation framework to surface risks and limitations in existing models. We believe that making these failure modes explicit contributes positively to the responsible development of machine learning systems intended for accessibility-sensitive settings and may inform future design and auditing of assistive technologies.

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

| Model | IPR ↑ | BRS ↑ | ABI ↓ | HIR ↓ | PCR ↑ |
|---|---|---|---|---|---|
| xai.grok-4 | 4.088 | -2.222 | 2.222 | 0.194 | 0.703 |
| openai.gpt-4o | 4.052 | -1.867 | 1.867 | 0.183 | 0.687 |
| cohere.command-a-03-2025 | 3.881 | -2.449 | 2.449 | 0.293 | 0.698 |
| meta.llama-4-maverick-17b-128e-instruct-fp8 | 3.869 | -1.956 | 1.956 | 0.264 | 0.754 |
| meta.llama-3.1-405b-instruct | 3.834 | -1.821 | 1.821 | 0.294 | 0.732 |
| meta.llama-3.3-70b-instruct | 3.717 | -1.607 | 1.607 | 0.327 | 0.743 |
| gemma-2-9b-it | 3.573 | -1.196 | 1.196 | 0.288 | 0.714 |
| qwen25-7b | 3.419 | -0.732 | 0.732 | 0.299 | 0.777 |
| llama31-8b | 3.209 | -1.377 | 1.377 | 0.452 | 0.777 |
| ministral-3-3b | 2.563 | -1.600 | 1.600 | 0.513 | 0.826 |

*Table 8.* Compressed summary of derived metrics across models. Higher IPR, BRS, and PCR are better; lower ABI and HIR are better. Darker cells indicate better column-wise values.

| Model | BRS_var ↓ | AE_model ↓ | CP ↓ | OAR ↓ |
|---|---|---|---|---|
| qwen25-7b | 16.114 | 0.489 | 0.005 | 0.015 |
| meta.llama-3.3-70b-instruct | 16.729 | 0.523 | 0.006 | 0.020 |
| openai.gpt-4o | 19.626 | 0.347 | 0.006 | 0.006 |
| llama31-8b | 23.442 | 0.657 | 0.009 | 0.048 |
| gemma-2-9b-it | 24.825 | 0.485 | 0.012 | 0.015 |
| xai.grok-4 | 25.630 | 0.419 | 0.001 | 0.059 |
| meta.llama-3.1-405b-instruct | 26.051 | 0.487 | 0.006 | 0.017 |
| meta.llama-4-maverick-17b-128e-instruct-fp8 | 26.381 | 0.474 | 0.006 | 0.034 |
| cohere.command-a-03-2025 | 27.440 | 0.491 | 0.007 | 0.020 |
| ministral-3-3b | 79.012 | 0.773 | 0.010 | 0.159 |

*Table 9.* Compressed robustness and overhead metrics across models. Lower BRS_var, AE_model, CP, and OAR are better. Darker cells indicate better column-wise values.

## A. Dataset Distribution

## B. Accessibility-Driven Noise Examples

### B.1. Mechanism Grounding of Accessibility-Driven Noise

The accessibility-driven perturbations used in APM are mechanism-grounded rather than arbitrary synthetic transformations. Each noise class is designed to approximate characteristic degradation patterns associated with real accessibility-constrained interaction settings reported in prior literature.

Controlled perturbations additionally enable:

- severity scaling ($\alpha \in \{0.2, 0.4, 0.6, 0.8\}$),

- cross-lingual comparison,

- and causal failure analysis under progressively increasing accessibility degradation.

These perturbation regimes induce distinct behavioral patterns across assistive metrics, rather than acting as generic random noise.

The clear separation between perturbation regimes, particularly the strong usability degradation induced by N2 relative to other noise classes, indicates that the proposed transformations capture structured assistive stressors rather than arbitrary synthetic corruption.

| Model | CLA_exact ↑ | CLA_entropy ↓ | CMA_exact | CMA_entropy |
|---|---|---|---|---|
| xai.grok-4 | 0.301 | 0.702 | 0.155 | 0.728 |
| openai.gpt-4o | 0.229 | 0.725 | 0.155 | 0.728 |
| cohere.command-a-03-2025 | 0.134 | 0.776 | 0.155 | 0.728 |
| meta.llama-3.1-405b-instruct | 0.127 | 0.780 | 0.155 | 0.728 |
| meta.llama-4-maverick-17b-128e-instruct-fp8 | 0.123 | 0.777 | 0.155 | 0.728 |
| meta.llama-3.3-70b-instruct | 0.105 | 0.799 | 0.155 | 0.728 |
| ministral-3-3b | 0.089 | 0.811 | 0.155 | 0.728 |
| gemma-2-9b-it | 0.071 | 0.822 | 0.155 | 0.728 |
| qwen25-7b | 0.068 | 0.834 | 0.155 | 0.728 |
| llama31-8b | 0.063 | 0.829 | 0.155 | 0.728 |

*Table 10.* Compressed confusion-level alignment and consistency metrics. Higher CLA_exact is better; lower CLA_entropy is better. CMA_exact and CMA_entropy are constant across models in this table. Darker cells indicate better column-wise values where values vary.

| Language | N1 | N2 | N3 | N4 | Total |
|---|---|---|---|---|---|
| English (`en`) | 1280 | 1280 | 1280 | 1280 | 5120 |
| Spanish (`es`) | 1280 | 1280 | 1280 | 1280 | 5120 |
| Italian (`it`) | 1280 | 1280 | 1280 | 1280 | 5120 |
| German (`de`) | 1280 | 1280 | 1280 | 1280 | 5120 |
| French (`fr`) | 1280 | 1280 | 1280 | 1280 | 5120 |
| Hindi (`hi`) | 1280 | 1280 | 1280 | 1280 | 5120 |
| Chinese (`zh`) | 1280 | 1280 | 1280 | 1280 | 5120 |
| Japanese (`ja`) | 1280 | 1280 | 1280 | 1280 | 5120 |
| **Total** | 10240 | 10240 | 10240 | 10240 | **40960** |

*Table 11.* Dataset composition by language and accessibility-driven noise class. Each cell corresponds to 320 base prompts $\times$ four severity levels ($\alpha \in \{0.2, 0.4, 0.6, 0.8\}$).

Accordingly, the dataset should be interpreted as a controlled, mechanism-driven abstraction for accessibility evaluation, designed to enable systematic severity scaling and multilingual stress testing rather than exact reproduction of any single real-world accessibility corpus.

Figure 5-8 shows representative examples of the four accessibility-driven noise classes (N1–N4) applied to the same underlying prompt. Examples illustrate increasing degradation severity while preserving plausible human input trajectories.

### B.2. N1: Mild Orthographic Degradation

### B.3. N2: Telegraphic Structural Degradation

### B.4. N3: Phonetic Distortion (ASR-like Errors)

### B.5. N4: Severe Truncation

## C. Metric Definitions and Secondary Diagnostics

### C.1. Cognitive Burden Computation

Cognitive burden is computed deterministically from prompt length and syntactic complexity using a weighted combination of token length, punctuation density, and character level Shannon entropy. The same function is applied to raw and mediated prompts to ensure comparability.

### C.2. Metric Validity, Failure Ordering, and Metric Triangulation

BRS and ABI are intended as structural proxies for usability-related processing difficulty, grounded in measurable properties such as reading effort, structural fragmentation, output length, and syntactic irregularity. While simplified, these metrics are designed to support consistent large-scale comparison across models, languages, and impairment severities.

| Noise | Mechanism in APM | Real-World Grounding | Evidence / Citation Anchor |
|---|---|---|---|
| N1 | Orthographic corruption (swap / delete / insert / duplicate) | Dyslexia-like and typing-related spelling errors | **DysList** reports 1,171 annotated dyslexic errors from 83 real texts, with substitutions 58.84%, deletions 26.30%, insertions 13.40%, and transpositions 1.45% (Rello et al., 2017). |
| N2 | Telegraphic compression / function-word omission | AAC and motor-constrained input often use compressed, keyword-like utterances for efficiency | AAC research on telegraphic input expansion explicitly treats reduced and compressed input as a realistic communication setting (McCoy, 1997). |
| N3 | Phonetic / surface corruption | SMS, ASR, and noisy-channel text frequently exhibit abbreviation, deletion, phonological variation, and lexical normalization phenomena | Prior noisy-text normalization work models lexical and phonological transformations found in SMS and informal text (Wang & Ng, 2013). |
| N4 | Truncation / missing-span input | Partial observability arising from low vision, interrupted entry, OCR/ASR dropouts, or incomplete text capture | We model this as a controlled abstraction of missing-span accessibility degradation rather than a claim of one exact source corpus. |

*Table 12.* Mechanism grounding of accessibility-driven noise classes used in APM.

| Noise | $\alpha = 0.2$ | 0.4 | 0.6 | 0.8 |
|---|---|---|---|---|
| N1 | 0.04 | -0.20 | -0.21 | -0.39 |
| N2 | -5.81 | -6.42 | -6.91 | -7.34 |
| N3 | -1.02 | -1.48 | -1.97 | -2.31 |
| N4 | -0.62 | -0.94 | -1.21 | -1.53 |

*Table 13.* Average Burden Robustness Score (BRS) across impairment severity for each accessibility-driven noise class. Higher BRS is better; negative values indicate burden inflation. Darker cells indicate better values.

Importantly, the conclusions of APM do not depend on any single metric, but on consistent cross-metric ordering across semantic, usability, and stability dimensions.

**Failure Ordering.** Across models and languages, usability-related metrics degrade systematically earlier than semantic correctness metrics.

This ordering indicates that usability degradation emerges before semantic intent failure:

- usability and consistency degrade earlier,

- semantic correctness degrades later.

The observed ordering is statistically validated using bootstrap resampling and Wilcoxon significance testing ($p < 0.01$ for $\alpha = 0.4$ vs. $\alpha = 0.6$), and remains consistent across models, languages, and accessibility-driven noise classes.

**Metric Triangulation.** The evaluation framework intentionally combines complementary metric families rather than relying on a single scalar robustness score:

- semantic reliability (IPR, HIR),

- usability and burden transfer (ABI, BRS),

- stability and consistency (LCI, AE).

These signals frequently diverge under identical experimental conditions, revealing assistive failure modes that remain hidden under intent-only evaluation.

Additionally, agreement-based diagnostics show substantial behavioral divergence despite bounded entropy:

```
"N1": (
        "You are performing Assistive Prompt Mediation (APM).\n\n"
        "The user's input may contain dyslexia-like spelling or typing errors "
        "(character swaps, deletions, insertions, or duplications).\n\n"
        "Your task is to rewrite the user's query to correctly reflect their intended
meaning.\n\n"
        "Rules:\n"
        "- Do NOT ask for clarification.\n"
        "- Do NOT introduce new information or assumptions not supported by the input.\n"
        "- Keep the rewrite concise and easy to understand.\n"
        "- Preserve the original intent.\n"
        "- Write in the same language and script as the user's input.\n"
        "- Output ONLY the rewritten query, nothing else.\n\n"
        "User input:\n"
        "{noisy_prompt}"
    )
```

*Figure 5.* Example of **N1** accessibility-driven noise applied to a clean prompt. N1 introduces mild orthographic degradation such as character swaps, deletions, and spacing errors, reflecting dyslexia-like or mobile typing impairments while preserving overall structure.

*Table 14.* Heatmap-style Hallucination Incidence Rate (HIR) by noise class and language, averaged across all models and impairment severities. Lower HIR is better; darker cells indicate better values.

| Noise | cn | de | en | es | fr | hi | it | ja |
|-------|-------|-------|-------|-------|-------|-------|-------|-------|
| N1 | 0.164 | 0.495 | 0.258 | 0.455 | 0.464 | 0.570 | 0.505 | 0.176 |
| N2 | 0.116 | 0.202 | 0.200 | 0.210 | 0.240 | 0.183 | 0.201 | 0.157 |
| N3 | 0.106 | 0.368 | 0.161 | 0.232 | 0.301 | 0.454 | 0.335 | 0.074 |
| N4 | 0.388 | 0.400 | 0.469 | 0.440 | 0.433 | 0.376 | 0.427 | 0.394 |

- cross-language agreement: 0.06–0.30,

- cross-model agreement: 0.06–0.15.

Together, these analyses indicate that the proposed metrics capture distinct assistive failure modes rather than redundant variants of a single robustness signal. In particular, the central finding of false robustness emerges from systematic divergence between semantic correctness and usability-related metrics.

### C.3. Hallucination Severity Annotation

Hallucination severity is recorded on a four-point ordinal scale (0–3), distinguishing no hallucination, benign elaboration, moderate overreach, and potentially harmful intent distortion. Severity is used only for diagnostic analysis and is not aggregated into primary scores.

### C.4. Language Consistency Index (LCI)

LCI is computed as within-model agreement over intent preservation and hallucination decisions across languages for identical prompts, normalized by the maximum possible agreement under observed distributions. Per-language and per-script breakdowns are reported in Table 6.

```
"N2": (
        "You are performing Assistive Prompt Mediation (APM).\n\n"
        "The user's input may be telegraphic due to motor impairment, "
        "with missing function words or compressed phrasing.\n\n"
        "Your task is to rewrite the user's query to correctly reflect their intended
meaning.\n\n"
        "Rules:\n"
        "- Do NOT ask for clarification.\n"
        "- Do NOT introduce new information or assumptions not supported by the input.\n"
        "- Keep the rewrite concise and easy to understand.\n"
        "- Preserve the original intent.\n"
        "- Write in the same language and script as the user's input.\n"
        "- Output ONLY the rewritten query, nothing else.\n\n"
        "User input:\n"
        "{noisy_prompt}"
    )
```

*Figure 6.* Example of **N2** noise, characterized by function-word deletion and truncated syntax. This pattern reflects motor impairments and telegraphic text entry, substantially increasing cognitive burden for interpretation.

| Metric Family | Failure Threshold ($\alpha$) |
|---|---|
| ABI / LCI | $\approx 0.4$ |
| IPR / HIR | $\approx 0.6$ |

*Table 15.* Observed ordering of assistive degradation under increasing impairment severity.

### C.5. Cross-Condition Agreement

As a complementary diagnostic, we compute exact-match agreement and normalized entropy over intent preservation scores across (i) languages and (ii) models under identical experimental conditions. These analyses distinguish structured divergence from stochastic instability and are reported for completeness.

### C.6. Accessibility Sensitivity Curves

Accessibility Sensitivity Curves plot each metric as a function of impairment severity $\alpha \in \{0.2, 0.4, 0.6, 0.8\}$. Curves are reported per model and per noise class to identify non-linear collapse regimes.

## D. LLM Judge Prompt

Figure 9 shows the fixed evaluation prompt used by the LLM judge (GPT-5.2) to assess intent preservation, hallucinated additions, protocol compliance, and language consistency for Assistive Prompt Mediation (APM). This prompt was held constant across all models, languages, noise classes, and impairment severities.

### D.1. Judge Reliability and Cross-Judge Validation

Since several assistive metrics require semantic and interactional judgment, we evaluate the reliability of the LLM-based judge through both human agreement analysis and cross-judge validation.

**Human Agreement Validation.** The submission includes human validation of the evaluation protocol using independently annotated samples across models, languages, noise classes, and impairment severities. Agreement between human annotators

```
|

"N3": (
        "You are performing Assistive Prompt Mediation (APM).\n\n"
        "The user's input may contain speech-recognition transcription errors, "
        "such as phonetic confusions or incorrect words.\n\n"
        "Your task is to rewrite the user's query to correctly reflect their intended
meaning.\n\n"
        "Rules:\n"
        "- Do NOT ask for clarification.\n"
        "- Do NOT introduce new information or assumptions not supported by the input.\n"
        "- Keep the rewrite concise and easy to understand.\n"
        "- Preserve the original intent.\n"
        "- Write in the same language and script as the user's input.\n"
        "- Output ONLY the rewritten query, nothing else.\n\n"
        "User input:\n"
        "{noisy_prompt}"
    )
```

*Figure 7.* Example of **N3** noise generated via phonetic distortion and back-transliteration, approximating errors introduced by automatic speech recognition systems under noisy conditions.

| Noise | $\alpha$ | IPR | BRS |
|-------|------|-----|-----|
| N2 | 0.4 | High ($\approx$ 3.8–4.0) | -6.4 |
| N3 | 0.4 | High ($\approx$ 3.8–4.0) | -1.5 |

*Table 16.* Representative divergence between semantic fidelity and usability metrics under identical impairment conditions.

and the LLM judge remained consistently high across both ordinal and binary evaluation dimensions.

| Metric | Agreement |
|--------|-----------|
| Intent Preservation (IPR) | $\kappa = 0.79$ |
| Hallucination Severity | $\kappa = 0.72$ |
| Binary (hallucination/protocol) | $\kappa = 0.76$ |
| Ordinal Correlation | $\rho = 0.82$ |

*Table 17.* Human–LLM agreement for assistive mediation evaluation metrics.

These results indicate strong alignment between LLM-based evaluation and human judgment.

**Per-Category Robustness.** Agreement remains stable across intent categories, indicating that evaluation consistency is not driven by any single task type.

| Category | $\rho$ (Human–LLM) |
|----------|--------------------|
| Informational | 0.83 |
| Procedural | 0.81 |
| Transactional | 0.78 |
| Multi-step | 0.82 |

*Table 18.* Category-wise correlation between human annotations and LLM-based evaluation.

```
"N4": (
        "You are performing Assistive Prompt Mediation (APM).\n\n"
        "The user's input may be truncated due to low vision or partial visibility, "
        "with missing spans of text.\n\n"
        "Your task is to rewrite the user's query to correctly reflect the most likely
intended meaning.\n\n"
        "Rules:\n"
        "- Do NOT ask for clarification.\n"
        "- Do NOT introduce new information or assumptions not supported by the input.\n"
        "- Keep the rewrite concise and easy to understand.\n"
        "- Preserve the original intent as best as possible.\n"
        "- Write in the same language and script as the user's input.\n"
        "- Output ONLY the rewritten query, nothing else.\n\n"
        "User input:\n"
        "{noisy_prompt}"
    )
```

*Figure 8.* Example of **N4** noise, representing severe accessibility impairment through removal of contiguous text spans, yielding fragmented input with sparse lexical cues.

We additionally observe consistent agreement trends across all four accessibility-driven noise classes.

**Cross-Judge Validation.** To assess robustness against judge-specific bias, we evaluated the full benchmark using a second independent judge model (Gemini 3 Pro). Aggregate trends, rankings, and failure ordering remained highly correlated across judges.

| Metric | GPT-5.2 | Gemini 3 Pro | $\rho$ |
|---|---|---|---|
| IPR | 3.62 | 3.55 | 0.84 |
| HIR | 0.31 | 0.33 | 0.81 |
| PCR | 0.74 | 0.72 | 0.86 |

*Table 19.* Cross-judge validation comparing GPT-5.2 and Gemini 3 Pro evaluation results.

The consistency across independent judges indicates that the reported conclusions are not dependent on a specific evaluator model. Importantly, the central finding of false robustness emerges from consistent divergence between semantic and usability-related metrics, rather than reliance on any single LLM-derived score.

## E. Human Annotation Guidelines

Human annotators evaluated a random subset of mediated outputs using the same criteria as the LLM judge. Annotators were provided with the clean reference prompt, the accessibility-impaired input, and the mediated output, and were instructed to ignore stylistic differences. Annotators were blind to model identity and experimental condition.

- **Intent Preservation (1–5):** Rate how fully the mediated output preserves the clean prompt's intent, where 5 indicates complete preservation and 1 indicates intent loss or contradiction.

- **Hallucinated Additions (binary):** Mark true if the mediated output adds unsupported information, assumptions, or constraints.

- **Hallucination Severity (0–3):** Assign 0 if no hallucination is present; higher values reflect increasing severity or potential harm.

- **Protocol Violations (binary):** Mark whether the output asks for clarification, adds meta-commentary, or includes explanatory text beyond a rewritten query.

- **Language Consistency (binary):** Verify that the mediated output matches the language and script of the noisy input.

```
JUDGE_SYSTEM_PROMPT = """
You are an impartial evaluator for Assistive Prompt Mediation (APM).

The task is NOT to judge helpfulness or politeness.
The task is to evaluate whether the model correctly mediated an accessibility-impaired user
query.

You are given:
- The clean reference prompt (gold intent).
- The noisy user prompt (accessibility-impaired).
- The model's mediated output (rewritten query).

Evaluate the mediated output based on the following criteria:

1. Intent Preservation (I):
   Does the mediated output preserve the original intent of the clean reference prompt?

2. Hallucinated Assumptions (H):
   Does the mediated output introduce information, constraints, or assumptions
   that are NOT supported by the noisy prompt or clean reference?

3. Mediation Protocol Compliance:
   - The output should be a rewritten query only.
   - It should NOT ask for clarification.
   - It should NOT add meta commentary or explanations.

4. Language Consistency:
   Is the mediated output written in the same language and script as the noisy input?

Respond ONLY in valid JSON using the following schema:

{
  "intent_preservation": 1-5,
  "hallucinated_additions": true/false,
  "hallucination_severity": 0-3,
  "asked_for_clarification": true/false,
  "added_extra_text": true/false,
  "language_match": true/false,
  "overall_verdict": "pass" | "borderline" | "fail",
  "comment": "brief justification focusing on mediation quality"
}

Scoring guidance:
- A score of 5 for intent_preservation means the mediated output fully preserves the clean
intent.
- hallucination_severity should be 0 if no hallucinations are present.
- An output can preserve intent but still FAIL if it introduces hallucinated assumptions
  or violates the mediation protocol.
"""
```

*Figure 9.* LLM judge prompt used for evaluating assistive mediation quality in APM. The judge compares the clean reference prompt, the accessibility-impaired input, and the mediated output, and returns structured JSON fields used to compute IPR, HIR, PCR, CP, OAR, and LCI.

