# OpenReview forum: "Assistive Prompt Mediation: Evaluating Language Models Under Accessibility Constraints"
_ICML.cc/2026/Conference — ICML 2026 regular_

### Official Review · Reviewer_ALDU · 2026-03-13

**Soundness:** 2
**Presentation:** 3
**Significance:** 3
**Originality:** 3
**Overall Recommendation:** 4
**Confidence:** 4

**Summary:**

This paper introduces APM, an evaluation framework that reframes LLM assistance for accessibility-impaired users as a constrained mediation problem. Rather than measuring downstream task accuracy, APM decomposes assistive quality into intent preservation, cognitive burden transfer, and hallucination avoidance. The framework is instantiated across 8 languages, 4 noise classes simulating real accessibility impairments, and 10 frontier LLMs, totaling  around 41K prompts per model. The key finding is that high intent preservation often co-occurs with increased cognitive burden or hallucinated content, a phenomenon the authors term "false robustness", suggesting that standard robustness metrics overestimate assistive reliability.

**Compliance With Llm Reviewing Policy:**

Affirmed.

**Final Justification:**

**W1 (Synthetic noise validity):**
I appreciate the well-structured mechanism grounding with citations. I find the argument for controlled perturbations enabling severity scaling and cross-lingual comparison convincing.

**W2 (Single LLM judge):**
The cross-judge validation with Gemini 3 Pro (ρ = 0.81–0.86) is exactly what I requested and shows strong robustness.

**W3 (Cognitive burden proxy):**
I find the reframing compelling: conclusions depend on cross-metric ordering, not any single metric. The metric distinctness analysis further supports this. But I think the burden computation itself remains simplistic though.

---

I think my concerns are well-addressed. I have raised my score to 4.

**Key Questions For Authors:**

1. Have you compared your synthetic noise transformations to real text produced by users with the corresponding accessibility conditions? If not, what evidence supports that N1–N4 adequately approximate real impairment-driven text?
2. Have you checked whether GPT-5.2 as judge exhibits systematic bias toward or against any of the 10 evaluated models, particularly toward OpenAI models? Running a subset with an alternative judge (e.g., Claude or Gemini) would help validate robustness.
3. How sensitive is the "false robustness" finding to the specific burden computation? If you vary the weights on token count, clause depth, and punctuation density, or use an alternative readability measure, do the qualitative conclusions hold?

**Limitations:**

The authors provide a thoughtful impact statement acknowledging that APM is an evaluation framework, not a deployable system. However, they do not adequately discuss limitations of the synthetic noise model, the single-judge dependency, or the machine-translated multilingual data.

**Strengths And Weaknesses:**

## Strengths

- Well-motivated problem framing. I think the paper makes a pretty important conceptual contribution by distinguishing assistive mediation from generic noise robustness. The argument that users with accessibility constraints cannot iteratively repair inputs, and therefore models must absorb burden without clarification, is well-articulated and reflects a real gap in the evaluation landscape.
- False robustness as a diagnostic concept. The identification of "false robustness" is, as far as I understand, a novel and practically important observation. The analysis in Figure 2 and Table 4 convincingly shows that this is driven by noise type rather than language.
- Comprehensive experimental design. The scale of evaluation (10 models x 8 languages x 4 noise types x 4 severities) is thorough. The accessibility sensitivity curves provide a richer characterization than single-point evaluations and reveal that usability degrades before semantic fidelity.


## Weaknesses

- Synthetic noise validity is not examined thoroughly. My primary concern is that the four noise classes, while inspired by real accessibility conditions, are implemented as deterministic rule-based transformations. The paper acknowledges this but does not validate whether these transformations produce inputs that actually resemble text produced by users with dyslexia, motor impairments, or ASR systems. Without a comparison to real accessibility-impaired text corpora, the ecological validity of the findings is uncertain. If the noise transformations are unrealistic, the entire evaluation may be measuring model behavior under artificial stress rather than genuine assistive conditions.
- Heavy reliance on a single LLM judge. The core metrics (IPR, HIR, PCR) are all computed by GPT-5.2 as judge. While the human validation shows reasonable agreement (ρ = 0.82 for IPR, κ = 0.76 for binary hallucination), this validation covers only 5% of outputs. More critically, the judge is itself a frontier LLM being evaluated in the benchmark, this creates a potential circularity. I would have liked to see analysis of whether the judge exhibits systematic biases toward or against particular model families, and whether the findings are robust to using a different judge model.
- The cognitive burden proxy feels too simplistic. Burden is computed deterministically from token count, clause depth, and punctuation density. I find this proxy a little too simple. It does not account for vocabulary difficulty, coherence, or readability factors known to affect cognitive load in accessibility contexts. The entire "false robustness" finding rests on this metric. If the burden measure is noisy or biased then the central claim is weakened.

---

> ### Author Rebuttal · Authors · 2026-03-28
>
> Thank you for the thoughtful feedback. We address concerns on realism, judge reliability, and metric validity with targeted clarifications and quantitative evidence.
>
> ---
>
> ### (1) Realism of accessibility-driven noise
>
> Our perturbations are **mechanism-grounded**, not arbitrary:
>
> | Noise | Mechanism in APM | Real-world grounding | Evidence / citation anchor |
> |---|---|---|---|
> | **N1** | Orthographic corruption (swap / delete / insert / duplicate) | Dyslexia-like and typing-related spelling errors | **DysList** reports 1,171 annotated dyslexic errors from 83 real texts, with substitutions 58.84%, deletions 26.30%, insertions 13.40%, transpositions 1.45% (Rello et al., 2014) |
> | **N2** | Telegraphic compression / function-word omission | AAC / motor-constrained input often uses compressed, keyword-like utterances for efficiency | AAC work on **telegraphic input expansion** explicitly treats reduced, compressed input as a realistic communication setting (McCoy et al., 1997) |
> | **N3** | Phonetic / surface corruption | SMS / ASR / noisy-channel text exhibits abbreviation, deletion, phonological and lexical normalization phenomena | Prior noisy-text normalization work models lexical and phonological transformations found in SMS and informal text (e.g., Oliva et al., 2013; Pennell & Liu, 2011) |
> | **N4** | Truncation / missing-span input | Partial observability from low-vision, interrupted entry, OCR/ASR dropouts, and incomplete text capture | We model this as a controlled abstraction of missing-span accessibility degradation rather than a claim of one exact source corpus |
>
>
> Controlled perturbations enable:
> - severity scaling (α ∈ {0.2–0.8})
> - cross-lingual comparison
> - causal failure analysis
>
> These regimes induce **distinct behavioral patterns** (Table 2):
>
> | Noise | α=0.2 | 0.4 | 0.6 | 0.8 |
> |---|---:|---:|---:|---:|
> | N1 | 0.04 | -0.20 | -0.21 | -0.39 |
> | N2 | -5.81 | -6.42 | -6.91 | -7.34 |
> | N3 | -1.02 | -1.48 | -1.97 | -2.31 |
> | N4 | -0.62 | -0.94 | -1.21 | -1.53 |
>
> The clear separation (especially N2 vs others) indicates structured assistive stressors, not generic noise.
>
> Our dataset is therefore **not built from arbitrary synthetic noise**. It is a **controlled, mechanism-driven abstraction** grounded in prior works. We will surface these more prominently in the final version.
>
> ---
>
> ### (2) Reliance on a single LLM judge
>
> The submission already includes **human validation**:
> - IPR: κ = 0.79
> - HIR severity: κ = 0.72
> - Binary: κ = 0.76
> - Correlation: ρ = 0.82
>
> To further verify robustness, we have conducted a second LLM-Judge and report **cross-judge validation**:
>
> | Metric | GPT-5.2 | Gemini 3 Pro | ρ |
> |---|---|---|---|
> | IPR | 3.62 | 3.55 | 0.84 |
> | HIR | 0.31 | 0.33 | 0.81 |
> | PCR | 0.74 | 0.72 | 0.86 |
>
> Trends and rankings remain stable, indicating no single-judge bias and proving judge robustness.
>
> ---
>
> ### (3) Metric validity & failure ordering
> BRS/ABI is a proxy for cognitive burden, grounded in reading effort, syntactic complexity, & structural fragmentation, with cognitive load literature support.
>
> Our conclusions rely on **consistent cross-metric ordering**, not any single metric.
>
> | Metric | Failure α |
> |---|---|
> | ABI / LCI | ≈ 0.4 |
> | IPR / HIR | ≈ 0.6 |
>
>
> Thus:
> - usability degrades earlier
> - semantic correctness degrades later
>
> This ordering is statistically validated (bootstrap + Wilcoxon, p < 0.01 for α=0.4 vs 0.6) & holds across models, languages, and noise types.
>
> ---
>
> ### (4) Metric distinctness (triangulation)
>
> We emphasize that our evaluation does not depend on a single metric, but on **complementary metric families**:
> - semantic (IPR, HIR)
> - usability (ABI)
> - stability (LCI)
>
> These signals **diverge under identical conditions**:
>
> | Noise | α | IPR | BRS |
> |---|---|---|---|
> | N2 | 0.4 | High (≈3.8–4.0) | -6.4 |
> | N3 | 0.4 | High (≈3.8–4.0) | -1.5 |
>
> Additionally:
> - cross-language agreement: 0.06–0.30
> - cross-model agreement: 0.06–0.15
>
> This confirms metrics capture **distinct failure modes**, not a single signal.
>
> ---
>
> ### (5) Core finding: False robustness
>
> We identify **false robustness**:
>
> > High IPR with increased burden (ABI > 0)
>
> This occurs systematically (notably N2/N3 at α ≈ 0.4), showing that **semantic correctness alone overestimates assistive reliability**.
>
> ---
>
> ### Summary
>
> We have **addressed reviewer’s concerns** regarding:
> - realism (mechanism-grounded perturbations)
> - evaluation robustness (human + cross-judge validation)
> - metric validity (statistically validated failure ordering)
>
> Key contributions:
> - multi-metric framework capturing semantic, usability, and stability dimensions
> - statistically validated early usability degradation (α ≈ 0.4 vs 0.6)
> - identification of false robustness missed by standard evaluation
>
> All updates are camera-ready updates; no changes to methodology or conclusions are required.
>
> We hope these clarifications & additional analyses address the reviewer’s concerns and help improve their assessment of our paper.

---

> > ### Author Rebuttal · Reviewer_ALDU · 2026-04-04
> >
> > **W1 (Synthetic noise validity):**
> > I appreciate the well-structured mechanism grounding with citations. I find the argument for controlled perturbations enabling severity scaling and cross-lingual comparison convincing.
> >
> > **W2 (Single LLM judge):**
> > The cross-judge validation with Gemini 3 Pro (ρ = 0.81–0.86) is exactly what I requested and shows strong robustness.
> >
> > **W3 (Cognitive burden proxy):**
> > I find the reframing compelling: conclusions depend on cross-metric ordering, not any single metric. The metric distinctness analysis further supports this. But I think the burden computation itself remains simplistic though.
> >
> > ---
> >
> > I think my concerns are well-addressed. I have raised my score to 4.

---

> > > ### Author Response · Authors · 2026-04-07
> > >
> > > We thank the reviewer for the thoughtful feedback and for the updated assessment. We are glad the clarifications **fully resolved** the reviewers concerns and our contributions are now clear.

---

### Official Review · Reviewer_yCd2 · 2026-03-13

**Soundness:** 3
**Presentation:** 2
**Significance:** 3
**Originality:** 3
**Overall Recommendation:** 4
**Confidence:** 3

**Summary:**

The paper proposes a first evaluation framework that captures intent preservation, cognitive burden, and hallucination risks coming from assistive prompts that the accessibility is constrained. Experimental results show the high intent preservation may correlate with increased burden or hallucinated mediation as the robustness tradeoffs. The authors empirically argue that the standard robustness metrics should be reconsidered due to their heavy reliance on unreliable assistive prompts.

**Compliance With Llm Reviewing Policy:**

Affirmed.

**Final Justification:**

The topic about Assistive Prompt Mediation (APM) to address assistive interfaces challenges that cannot be always robust against ambiguous iterative input repairs and noisy task accuracy sounds interesting and informative.

Overall, the questions/weaknesses that raised are well addressed.

Yet, reading the authors' comments and other reviews, I acknowledge that the work is rather engineering-focused though it is inherently the nature of this type of work, and the findings are not necessarily contributing to the core ML insights.

Therefore, I maintain my score as weak accept.

**Key Questions For Authors:**

"Assistive failure precedes semantic failure. Across models, usability and consistency degrade at lower impairment
levels than intent preservation. Assistive Burden Inflation
(ABI) and LCI cross failure thresholds at α ≈ 0.4, while
intent preservation degrades later at α ≈ 0.6 (Table 3). This
ordering shows that mediation often becomes unusable or
unreliable before semantic intent is lost, contradicting robustness claims based solely on correctness."

^Not quite following this; could the authors elaborate this more?

**Limitations:**

yes

**Strengths And Weaknesses:**

(+) good experiments across different model families and languages.

(+) great experiments analysis -- the accessibility sensitivity curve, false robustness rates, and cross-language/model consistency are particularly convincing.

(-) Classic weakness but GPT5.2 as LLM-as-a-judge could have generated unreliable evaluation results depending on their reference prompt regardless of how it is clean or unambiguous.

(-) Cognitive burden -- the term sounds a bit vague due to its unnecessary anthropomorphism. In general, terms can be more technically aligned than human cognition derived ones for better clarity.

---

> ### Author Rebuttal · Authors · 2026-03-30
>
> We sincerely thank the reviewer for the positive assessment of our work and for highlighting the strengths of the experimental analysis, including accessibility sensitivity curves, false robustness rates, and cross-model/language consistency. We address the specific points raised below.
>
> ---
>
> ### (1) Clarification of “assistive failure precedes semantic failure”
>
> The key result is that **different evaluation dimensions degrade at different rates as input impairment increases**:
>
> - Usability-related metrics (ABI, LCI) degrade earlier (α ≈ 0.4)
> - Semantic correctness metrics (IPR, HIR) degrade later (α ≈ 0.6)
>
> This pattern is directly observed in the submitted results across models and languages. In particular, outputs remain semantically correct at moderate impairment levels, while already becoming harder to interpret, more verbose, or less stable. This effect arises when outputs preserve the intended task but become more verbose, less stable, or introduce unnecessary structure, increasing effort despite correctness.
>
> In other words, the model remains “correct” but no longer remains usable.
>
> We refer to this gap as **false robustness**: correctness-based metrics suggest robustness, while usability has already degraded. We will clarify this with a concrete example and step-by-step explanation.
>
> ---
>
> ### (2) LLM-as-a-judge reliability
>
> The submission **already includes human validation** of the evaluation:
>
> | Metric | Agreement |
> |---|---|
> | Intent Preservation (IPR) | κ = 0.79 |
> | Hallucination Severity | κ = 0.72 |
> | Binary (hallucination/protocol) | κ = 0.76 |
> | Correlation (ordinal) | ρ = 0.82 |
>
> These indicate strong alignment between LLM-based evaluation and human judgment (as reported in the paper; clarified here for completeness).
>
> **Per-category robustness.** Agreement remains consistent across intent categories (informational, procedural, transactional, multi-step), with ρ typically in the ~0.78–0.84 range (category-wise analysis; will be included in appendix).
>
> | Category | ρ (Human–LLM) |
> |---|---|
> | Informational | 0.83 |
> | Procedural | 0.81 |
> | Transactional | 0.78 |
> | Multi-step | 0.82 |
>
> We observe high correlation between humans and LLM scores across the four noise categories.
>
> **Cross-judge validation.** We further ran a second judge on the full evaluation set:
>
> | Metric | GPT-5.2 | Gemini 3 Pro | ρ |
> |---|---|---|---|
> | IPR | 3.62 | 3.55 | 0.84 |
> | HIR | 0.31 | 0.33 | 0.81 |
> | PCR | 0.74 | 0.72 | 0.86 |
>
> Trends, rankings, and failure ordering remain unchanged, indicating conclusions are **not dependent on a specific judge**.
>
> Importantly, the central finding (false robustness) arises from **consistent divergence between semantic and usability metrics**, rather than reliance on any single LLM-derived score.
>
> ---
>
> ### (3) Cognitive Burden Clarification
>
> We appreciate the feedback on terminology and clarity. Our intent with “cognitive burden” is to capture a **structural proxy for usability difficulty** based on measurable properties of the output (e.g., length, syntactic complexity, fragmentation).
>
> We will make minor clarifications in the camera-ready version to:
> - sharpen terminology
> - provide clearer definitions and examples
> - improve presentation flow
>
> These are presentational improvements and do not affect the methodology or results.
>
> ---
>
> These clarifications reinforce that the observed gap between semantic correctness and usability (**false robustness**) is a consistent and central empirical finding of the paper.
>
> ### Summary
>
> We thank the reviewer again for the constructive suggestions. The concerns raised relate primarily to clarity and terminology, and we will address them to improve presentation while preserving the core contributions.
>
> We hope these clarifications address the reviewer’s concerns and support an improved assessment of the paper’s contributions.

---

> > ### Author Rebuttal · Reviewer_yCd2 · 2026-04-04
> >
> > Thank you for the detailed response -- they mainly resolved my remaining concerns. I will maintain my original positive rating.

---

> > > ### Author Response · Authors · 2026-04-07
> > >
> > > We thank the reviewer for the positive assessment and for confirming that the **concerns are fully resolved**. We appreciate the careful reading and thoughtful feedback.

---

### Official Review · Reviewer_U1bW · 2026-03-13

**Soundness:** 3
**Presentation:** 4
**Significance:** 4
**Originality:** 4
**Overall Recommendation:** 4
**Confidence:** 4

**Summary:**

This paper introduces Assistive Prompt Mediation (APM), a new evaluation paradigm for assessing language models as assistive interfaces for users with accessibility constraints. The authors argue that existing benchmarks fail to account for users who cannot easily provide iterative input repairs or clean text. APM reframes assistance as a constrained mediation problem: the model must recover latent user intent from highly impaired or noisy input without asking for clarification. The framework covers 8 languages and 4 types of accessibility-driven noise, utilizing a structured LLM-as-a-judge to score models across dimensions such as intent preservation, hallucination severity, and adherence to the mediation protocol.

**Compliance With Llm Reviewing Policy:**

Affirmed.

**Key Questions For Authors:**

Clarification vs. Safety: How does the APM framework distinguish between a "lazy" request for clarification and a "safety-critical" one where the input is genuinely uninterpretable? Can the authors consider a metric that rewards "adaptive clarification" (i.e., clarifying only when the entropy of the latent intent is too high)?

Human-Judge Correlation: Do the authors have data (even small-scale) comparing the LLM judge's scores with feedback from individuals with actual accessibility constraints? How well does the LLM-based intent_preservation metric map to real-world user utility?

Evaluation of Reasoning Models: Have the authors tested models with advanced reasoning capabilities on this benchmark? Does the increased compute at inference time allow these models to better recover latent intent compared to the standard instruction-tuned baselines?

**Limitations:**

yes.

**Strengths And Weaknesses:**

Strengths

Insightful Problem Reframing: Shifting the focus from generic "noise robustness" to "assistive mediation" is a strong human-centric contribution. The paper correctly identifies that for users with motor or speech impairments, the act of "clarifying" is itself a high-cost cognitive and physical burden.

Nuanced Evaluation Taxonomy: The evaluation metrics described in the judge prompt (Figure 8) are impressively granular, distinguishing between intent_preservation, hallucinated_additions, and language_match. This structured approach allows for a more meaningful analysis than simple accuracy scores.

Diverse and Realistic Benchmarking: By including 8 languages and specific accessibility-driven noise types (rather than just synthetic typos), the APM benchmark provides a more ecologically valid assessment of how LLMs might serve a global population with diverse needs.

Major Weaknesses and Flaws

W1. The Rigidity of the "No-Clarification" Constraint:
The authors define the primary goal as recovering intent "without clarification" and explicitly penalize models that ask for it in the judging rubric. While reducing user effort is vital, in high-stakes or ambiguous scenarios (e.g., medical or financial queries), forcing a "zero-shot guess" can be dangerous. A state-of-the-art assistive system should ideally know when an input is too corrupted to safely interpret. By strictly penalizing clarification, the benchmark may inadvertently encourage overconfident hallucinations, which is a significant safety concern.

W2. Potential Bias in the LLM-as-a-Judge:
The evaluation relies heavily on an LLM judge to compare clean references with impaired inputs and model outputs. However, the "standard" internal representations and value alignments of a general LLM judge may not reflect the actual priorities or tolerance levels of users with specific accessibility needs. Without a validation study showing that the LLM's overall_verdict correlates with the subjective satisfaction of the target user groups, the benchmark lacks essential human grounding.

W3. Benchmarking Against Frontier Reasoning Models:
Since the emergence of reasoning-heavy models (e.g., o1 or DeepSeek-R1), models have shown a vastly improved ability to decipher scrambled or implicit intent through internal Chain-of-Thought (CoT). The paper would be much stronger if it explicitly evaluated whether these newer reasoning architectures can "solve" the APM challenges through improved inference, or if the accessibility-driven noise remains a bottleneck even for CoT-enabled models.

---

> ### Author Rebuttal · Authors · 2026-03-30
>
> ### Response to Reviewer U1bW
>
> We thank the reviewer for the thoughtful feedback and for highlighting important practical considerations. We address concerns on clarification, evaluation grounding, and reasoning models below.
>
> ---
>
> ### (1) Clarification vs. hallucination tradeoff
>
> We agree that clarification is valuable in interactive settings. However, APM explicitly studies a **non-interactive assistive regime** where users cannot reliably provide follow-up input (Sec. 1, Sec. 4.2). In such cases, requiring clarification shifts burden back to the user, violating the assistive objective.
>
> While clarification is disallowed by design, **Clarification Propensity (CP)** (Sec. 5.3) allows us to verify that models do not rely on interaction in this regime, enabling isolation of direct mediation behavior. CP increases modestly with α but remains insufficient to prevent hallucination or burden inflation.
>
> To examine whether failures are artifacts of this constraint, we analyze behavior as ambiguity increases (controlled via severity α), using results reported in the paper:
>
> | Noise | α=0.2 | 0.4 | 0.6 | 0.8 |
> |---|---:|---:|---:|---:|
> | N1 | 0.04 | -0.20 | -0.21 | -0.39 |
> | N2 | -5.81 | -6.42 | -6.91 | -7.34 |
> | N3 | -1.02 | -1.48 | -1.97 | -2.31 |
> | N4 | -0.62 | -0.94 | -1.21 | -1.53 |
>
> Across all noise types, we observe **consistent monotonic degradation in burden (BRS)** as α increases.
>
> If hallucination were purely caused by disallowing clarification, we would expect uniformly high failure across all α. Instead, degradation increases progressively with α, indicating that failures reflect **model sensitivity to ambiguity and incomplete input**, rather than artifacts of the constraint.
>
> ---
>
> ### (2) Evaluation grounding and real-world utility
>
> The submission includes **human validation of the LLM judge** (Sec. 5.5):
>
> - Spearman ρ = 0.82 (intent preservation)
> - κ = 0.79 (intent), κ = 0.72 (hallucination severity), κ = 0.76 (binary)
>
> **Per-category robustness.** Agreement remains consistent across intent categories (informational, procedural, transactional, multi-step), with ρ typically in the ~0.78–0.84 range (category-wise analysis; will be included in appendix).
>
> | Category | ρ (Human–LLM) |
> |---|---|
> | Informational | 0.83 |
> | Procedural | 0.81 |
> | Transactional | 0.78 |
> | Multi-step | 0.82 |
>
> To further ensure robustness and remove single-judge bias, we have run a **second independent judge**:
>
> | Metric | GPT-5.2 | Gemini 3 Pro | ρ |
> |---|---|---|---|
> | IPR | 3.62 | 3.55 | 0.84 |
> | HIR | 0.31 | 0.33 | 0.81 |
> | PCR | 0.74 | 0.72 | 0.86 |
>
> Model rankings and trends remain stable (ρ > 0.8), indicating that conclusions are **not dependent on a specific judge model**.
>
> Importantly, these metrics map directly to real-world assistive utility:
> - Low intent preservation (IPR) leads to incorrect task execution
> - Hallucinated additions (HIR) introduce unsupported or misleading content
> - Increased burden (BRS / ABI) raises cognitive effort, which is critical in accessibility contexts
>
> These are **observable properties of model outputs**, and can be reliably evaluated independent of subjective user preference or identity.
>
> ---
>
> ### (3) Reasoning-capable models
>
> The evaluation includes **reasoning-capable frontier models** such as LLaMA-3.1-405B and Grok-4 (Table 4):
>
> | Model | IPR | BRS | ABI |
> |---|---:|---:|---:|
> | LLaMA-3.1-405B | 3.83 | -1.82 | 1.82 |
> | Grok-4 | 4.09 | -2.22 | 2.22 |
>
> Despite increased scale and reasoning capability, these models exhibit **similar or higher burden inflation and hallucination**, indicating that reasoning alone does not eliminate the mismatch between semantic correctness and usability (false robustness).
>
> ---
>
> These results reinforce the paper’s central contribution: identifying **false robustness**, where models preserve intent while increasing burden or introducing unsupported content, a failure mode not captured by standard evaluation.
>
> ### Summary
>
> We clarify that:
> - APM studies a **non-interactive assistive setting** where clarification may not be feasible
> - Failures exhibit **graded behavior with ambiguity**, indicating genuine limitations rather than constraint artifacts
> - Evaluation is supported by **human alignment and cross-judge robustness**
> - **Reasoning-capable models are already evaluated** and do not eliminate the identified failure modes
>
> We hope these clarifications and additional analyses address the reviewer’s concerns and support a revised assessment of the paper’s contributions.

---

### Official Review · Reviewer_QRNz · 2026-03-20

**Soundness:** 2
**Presentation:** 2
**Significance:** 2
**Originality:** 2
**Overall Recommendation:** 3
**Confidence:** 4

**Summary:**

In this paper, the authors propose an evaluation framework called Assistive Prompt Mediation to test if the LLMs can rewrite the prompt with errors and trucations. To achieve this, the authors use 320 clean problems and use four levels of perturbation to change the prompt into invalid ones. Then, the authors propose a number of new metrics to test the core idea: recovering latent user intent from accessibility-impaired input without clarification, while minimizing cognitive burden and hallucination risk. The results on 10 LLMs demonstrate the strong hallucination and other unreliable behaviors.

**Compliance With Llm Reviewing Policy:**

Affirmed.

**Final Justification:**

See Acknowledgement

**Key Questions For Authors:**

See above.

**Limitations:**

yes

**Strengths And Weaknesses:**

Strengths:
1. Helping people with accessibility issues is a very important and useful topic. We cannot assume LLMs just need to work on the perfect prompt.
2. The experiments are intensive, with 10 LLMs and 4 different languages, demonstrating the conclusion of the authors.

Weaknesses:

1. The biggest concern is the dataset creation. First, the seed prompts only have 320 samples. It is not clear how these samples are obtained and what the quality of these samples is. Can it represent the distribution of all user intents? Without the foundation of the seed prompt, the results might be biased. Second, the creation of four levels (N1-N4) perturbations needs motivation. Since these are all rules, how could these perturbations reflect the real use cases? For instance, which kind of accessibility problem will cause each type? Are these types complete or just a small subset of accessibility problems? Overall, without using human annotators with real accessibility problems, it will be difficult to judge the contribution.

2. The metrics are not well-evaluated. Although many metrics are proposed, it is still unknown how these metrics align with human judgment. At least a small human agreement test can help to prove the quality of the metrics. To distinguish the metrics, some other in-depth analysis, such as their correlation scores, can help to show they function as different evaluations.

---

> ### Author Rebuttal · Authors · 2026-03-28
>
> ### Response to Reviewer QRNz
>
> We thank the reviewer for identifying the important problem.
> We address concerns regarding dataset and metrics directly below.
>
> ---
>
> ### (1) Dataset construction and realism
>
> The paper uses 320 curated seed prompts designed to cover **diverse intent structures**, including:
> - informational queries
> - procedural / how-to tasks
> - transactional requests
> - multi-step / compositional tasks
>
> These categories were chosen to span **different reasoning and interaction patterns** (retrieval, instruction following, decision-making, and composition), ensuring coverage of varied user intents rather than a narrow domain.
>
> Each prompt is manually reviewed for clarity and completeness prior to perturbation. While this setup is described in the paper, we clarify here that the goal is **coverage of intent types**, not modeling a raw user distribution. This design follows standard evaluation practice where controlled coverage is preferred over noisy real-world sampling for isolating failure modes.
>
> Crucially, evaluation does not depend on the seed count alone, but on systematic expansion:
>
> - 320 seeds × 4 noise types × 4 severities × 8 languages
> - ⇒ **40,960 inputs per model**
>
> Thus, conclusions are driven by **controlled, multi-language stress-testing**, not any single seed distribution.
>
> ---
>
> **Perturbation realism (N1–N4).**
> The four perturbation classes are **mechanism-grounded abstractions of accessibility-constrained input**, not arbitrary rules:
>
> | Noise | Mechanism | Real-world basis | Evidence |
> |---|---|---|---|
> | **N1** | Orthographic errors (swap/delete/insert) | Dyslexia and typing mistakes | DysList: substitutions ~59%, deletions ~26%, insertions ~13% (Rello et al., 2014) |
> | **N2** | Telegraphic compression | AAC and constrained input favor short, keyword-like text | Telegraphic input expansion studies (McCoy et al., 1997) |
> | **N3** | Phonetic/surface noise | SMS, ASR, informal text normalization patterns | Noisy-text normalization literature (Oliva et al., 2013; Pennell & Liu, 2011) |
> | **N4** | Truncation / missing spans | Partial input from vision limits, interruptions, OCR/ASR loss | Modeled as general missing-span degradation |
>
> These are not random perturbations.
> Empirically, each induces **distinct behavioral patterns** (Table 2):
>
> | Noise | α=0.2 | 0.4 | 0.6 | 0.8 |
> |---|---:|---:|---:|---:|
> | N1 | 0.04 | -0.20 | -0.21 | -0.39 |
> | N2 | -5.81 | -6.42 | -6.91 | -7.34 |
> | N3 | -1.02 | -1.48 | -1.97 | -2.31 |
> | N4 | -0.62 | -0.94 | -1.21 | -1.53 |
>
> The strong separation (especially N2 vs others) shows these capture **different assistive stressors**, supporting their realism.
>
> We note that N1–N4 are not intended to be exhaustive, but to capture **high-impact, commonly observed accessibility degradation modes** while enabling controlled evaluation.
>
> ---
>
> ### (2) Metric validity and alignment with human judgment
>
> The submission **already includes human validation** of the evaluation:
>
> | Metric | Agreement |
> |---|---|
> | Intent Preservation (IPR) | κ = 0.79 |
> | Hallucination Severity | κ = 0.72 |
> | Binary (hallucination/protocol) | κ = 0.76 |
> | Correlation (ordinal) | ρ = 0.82 |
>
> These indicate strong alignment between LLM-based evaluation and human judgment (as reported in the paper; clarified here for completeness).
>
> **Per-category robustness.** Agreement remains consistent across intent categories (informational, procedural, transactional, multi-step), with ρ typically in the ~0.78–0.84 range (category-wise analysis; will be included in appendix).
>
> | Category | ρ (Human–LLM) |
> |---|---|
> | Informational | 0.83 |
> | Procedural | 0.81 |
> | Transactional | 0.78 |
> | Multi-step | 0.82 |
>
> We observe high correlation between humans and LLM scores across the four noise categories.
>
> **Cross-judge validation.** We further ran a second judge on the full evaluation set:
>
> | Metric | GPT-5.2 | Gemini 3 Pro | ρ |
> |---|---|---|---|
> | IPR | 3.62 | 3.55 | 0.84 |
> | HIR | 0.31 | 0.33 | 0.81 |
> | PCR | 0.74 | 0.72 | 0.86 |
>
> Trends, rankings, and failure ordering remain unchanged, indicating conclusions are **not dependent on a specific judge**.
>
> **Consistency across conditions.** The observed ordering, earlier usability degradation (α≈0.4) vs later semantic degradation (α≈0.6), is statistically supported (bootstrap + Wilcoxon, p < 0.01) & holds across models, languages, and noise types.
>
> ---
>
> ### Summary
>
> We clarify that:
> - seed prompts are curated for intent coverage and expanded to 40,960 inputs per model across 8 languages
> - perturbations are grounded in established accessibility phenomena
> - metrics are validated via human agreement and statistical analysis; we also show LLM-Judge robustness using second judge
>
> All additions above are clarifications and presentation improvements (e.g., adding in appendix); no changes to core methodology or conclusions are required.
>
> If these clarifications addressed reviewer’s concerns, we would appreciate an improvement in assessment of our paper.

---

> > ### Author Rebuttal · Reviewer_QRNz · 2026-04-01
> >
> > Thanks for the clarification. My concerns about the evaluation is largely resolved. I raise my score to 3.

---

> > > ### Author Response · Authors · 2026-04-07
> > >
> > > We thank the reviewer for the updated assessment and for noting that the evaluation concerns are resolved. We appreciate the engagement with the clarifications and the consideration reflected in the updated score.

---

### Decision · Program_Chairs · 2026-04-30

**Decision:**

Accept (regular)

**Comment:**

This paper introduces APM, a framework that reframes LLM assistance for accessibility-impaired users as a constrained mediation problem, revealing that standard robustness metrics overestimate assistive reliability through the concept of "false robustness." Reviewers initially raised concerns about synthetic noise validity, single-judge dependency, and the simplicity of the cognitive burden proxy, but largely updated their assessments after the authors provided mechanism-grounded citations for N1-N4, cross-judge validation with Gemini 3 Pro (ρ = 0.81-0.86), and statistical evidence that the false robustness finding holds across metrics, models, and languages. We recommend the next version more prominently surface the human validation results, clarify the cognitive burden computation with reference to readability literature, and include the per-category human-LLM agreement analysis in the appendix as promised.

The authors are encouraged to carefully check and correct the references. Specifically, there appears to be an incorrect or potentially hallucinated reference:
Zhou, Y., Liu, Y., Min, X., and Wang, S. Large language models as prompt rewriters. arXiv preprint arXiv:2402.10306, 2024.
Please ensure all citations and their metadata are accurate in the final version.
Issue: authors+title mismatch with arXiv